# Effects of CB2 Receptor Modulation on Macrophage Polarization in Pediatric Celiac Disease

**DOI:** 10.3390/biomedicines10040874

**Published:** 2022-04-09

**Authors:** Chiara Tortora, Alessandra Di Paola, Maura Argenziano, Mara Creoli, Maria Maddalena Marrapodi, Sabrina Cenni, Carlo Tolone, Francesca Rossi, Caterina Strisciuglio

**Affiliations:** 1Department of Woman, Child and General and Specialist Surgery, University of Campania “Luigi Vanvitelli”, Via L. De Crecchio 4, 80138 Naples, Italy; chiara.tortora@unicampania.it (C.T.); alessandra.dipaola@unicampania.it (A.D.P.); maurargenziano@gmail.com (M.A.); mariamaddalena.marrapodi@studenti.unicampania.it (M.M.M.); carlo.tolone@unicampania.it (C.T.); caterina.strisciuglio@unicampania.it (C.S.); 2Department of Experimental Medicine, University of Campania “Luigi Vanvitelli”, Via S. Maria di Costantinopoli 16, 80138 Naples, Italy; mara.creoli@unicampania.it (M.C.); sabrina.cenni@unicampania.it (S.C.)

**Keywords:** celiac disease, inflammation, intestinal barrier damage, macrophages, M1, M2, macrophage polarization, Caco-2 cells, CB2 receptor

## Abstract

Celiac Disease (CD) represents an autoimmune disorder triggered by the exposure to gluten in genetically susceptible individuals. Recent studies suggest the involvement of macrophages in CD pathogenesis. Macrophages are immune cells, present as pro-inflammatory classically activated macrophages (M1) or as anti-inflammatory alternatively activated macrophages (M2). The Cannabinoid Receptor 2 (CB2) has important anti-inflammatory and immunoregulatory properties. We previously demonstrated that a common CB2 functional variant, Q63R, causing CB2 reduced function, is associated with several inflammatory and autoimmune diseases The first aim of this study was to investigate the phenotype of macrophages isolated from peripheral blood of CD patients and CB2 expression. The second aim was to evaluate the effects of CB2 pharmacological modulation on CD macrophage polarization. Moreover, by an in vitro model of “immunocompetent gut” we investigated the role of CD macrophages in inducing intestinal barrier damage and the possibility to restore its functionality modulating their polarization. We found an increased expression of M1 macrophages and a CB2 reduced expression. We also demonstrated CD M1 macrophages in inducing the typical mucosal barrier damage of CD. CB2 stimulation switches macrophage polarization towards the anti-inflammatory M2 phenotype thus reducing inflammation but also limiting the epithelial dysfunction. Therefore, we suggest CB2 receptor as a possible novel therapeutic target for CD by regulating macrophages polarization and by preventing mucosal barrier damage.

## 1. Introduction

Celiac Disease (CD) represents an autoimmune disorder triggered by the exposure to gluten in genetically susceptible individuals, [1]. The main factors which concur to CD onset are genetic predisposition, altered intestinal epithelial barrier and chronic inflammatory immune response to gliadin [2,3]. Since the gluten-free diet (GFD) remains the only treatment for CD [4], the identification of novel therapies, alternative or complementary to GFD, is necessary. Recent studies suggest that macrophages contribute to CD pathogenesis [5,6]. Several studies report that gliadin actives both intestinal macrophages and circulating monocytes from which they are derived [7,8,9,10,11,12]. Recently, a macrophage-initiating granulomatous condition has been associated with CD [13]. Macrophages can exhibit a variety of phenotypes classified into classically (M1) and alternatively (M2) activated macrophages [14,15]. M1 macrophages have pro-inflammatory, antitumor and anti-microbial properties and release pro-inflammatory cytokines, such as TNF-α, Interleukin-6 (IL-6), Interleukin-1β (IL-1β) and Nitric Oxide Synthase [16,17] and are characterized by the expression of chemokine receptor 7 (C-C chemokine receptor type 7, CCR7) [18] and divalent metal transporter (DMT1) [19]. Conversely, M2 macrophages exert anti-inflammatory and immunosuppressive activities, releasing anti-inflammatory cytokines, such as IL-10 [20]. M2 macrophages are characterized by the expression of the mannose receptor C type 1 (MRC1) also known as CD206 [21]. Several members of the STAT family are involved in regulating macrophage functional status [22]. STAT6 signaling induces the anti-inflammatory phenotype of macrophages [23]. Furthermore, macrophages have a role in regulating iron homeostasis [24]. M1 macrophages are characterized by high iron content which promotes inflammation. In contrast, M2 macrophages are involved in iron release. In response to systemic iron requirements, iron release from macrophages into plasma is regulated by hepcidin [25]. During inflammation, high levels of IL-6 and other pro-inflammatory cytokines increase hepcidin synthesis which acts by binding the only iron exporter ferroportin 1 (FPN-1), causing iron sequestration in macrophages [26]. Macrophage polarization could change according to the inflammatory pattern of their microenvironment [27]. Therefore, novel possible therapeutic strategies for CD could aim to induce a macrophage switch from M1 pro-inflammatory phenotype to M2 anti-inflammatory one, reverting the chronic inflammatory immune response that characterized the disease. The Cannabinoid Receptor 2 (CB2) has important anti-inflammatory and immunoregulatory properties [28]. It suppresses immune cell activation modulating T helper cells and inhibits pro-inflammatory cytokine production [29,30]. Moreover, a common CB2 functional variant, Q63R, causing CB2 reduced function, has been associated with several inflammatory and autoimmune diseases [31,32,33] In this study, we isolated macrophages from the peripheral blood of CD patients to investigate CB2 expression and macrophage phenotype and to evaluate the effects of CB2 pharmacological modulation on CD macrophage polarization. Then, by an in vitro model of the “immunocompetent gut” [34], constituted by CD macrophages and the human epithelial cell line, Caco-2, we investigated the possible role of CD macrophages in inducing intestinal barrier damage and the possibility to restore its functionality modulating their polarization

## 2. Materials and Methods

### 2.1. Patients

This study was conducted using macrophages isolated from the peripheral blood mononucleated cells (PBMCs) of 10 CD children (median age 10 ± 4 years; 40% males) at diagnosis and 10 healthy controls (CTR) (median age 12 ± 5 years; 50% males) matching in age and gender. CTR were subjects affected by functional GI disorders, who performed blood analysis to exclude any organic diseases and an inflammatory condition. Patients and CTR were enrolled at the Department of Women, Child and General and Specialist Surgery of University of Campania Luigi Vanvitelli between May 2021 and December 2021. The inclusion criteria for CD patients were confirmed diagnosis of CD according to Espghan guidelines [35], absence of associated autoimmune diseases, absence of iron deficiency and no assumption of nutritional integration; for the healthy controls, the inclusion criteria were absence of diagnosed chronic disease and gastrointestinal disease. Clinical characteristics of CTR and CD subjects are reported in Table 1. All procedures executed in this study are in agreement with the Helsinki Declaration of Principles, the Italian National Legislation and the Ethics Committee of the University of Campania Luigi Vanvitelli, which formally approved the study (Identification code 499, 12 September 2017). Written informed consent was obtained from participants’ parents and from patients themselves if older than 10 years.

### 2.2. Macrophages Cell Cultures

Macrophages were isolated from peripheral blood mononuclear cells (PBMCs). PBMCs were isolated by density gradient centrifugation (Ficoll 1.077 g/mL),then diluted at 1 × 10^6^ cells/mL in α-Minimal Essential Medium (α-MEM) (Lonza, Verviers, Belgium) and plated in a 24-well cell culture multiwell. α-MEM was supplemented with 10% fetal bovine serum (FBS) (Euroclone, Siziano, Italy), 100 IU/mL penicillin and 100 g/mL streptomycin and L-glutamine (Gibco Limited, Uxbridge, UK) The PBMCs were cultured for 15 days in the presence of 25 ng/mL recombinant human macrophage colony-stimulating factor (rh-MCSF) (Peprotech, London, UK) to obtain mature human macrophages. Culture medium was replaced twice a week. Cells were cultured at 37 °C in a humidified atmosphere with 5% CO_2_. After 15 days of differentiation, cells were harvested for protein extraction, and cell culture supernatants were collected to analyze iron concentration (Iron Assay) and to analyze pro/anti-inflammatory cytokine release with an enzyme-linked immunosorbent assay (ELISA).

### 2.3. Caco-2 Cells Culture

Caco-2 cell line, a human colorectal adenocarcinoma cell line, was purchased from ATCC, accession number HTB-37. The cells were cultured in minimum essential medium (MEM) supplemented with 10% FBS, 1% non-essential amino acids and 1% antibiotics (100 U/mL penicillin and 100 µg/mL streptomycin). The culture medium was replaced twice a week. After reaching 80% confluence, Caco-2 were split, re-plated for expansion and harvested until passage 10. 

### 2.4. Macrophages and Caco-2 Cells Coculture in a Transwell System

Transwell experiments were performed as previously described [36]. Briefly, Caco2 cells were grown for 14–20 days to confluency in the upper chamber of a 3 μm pore transwell insert (Falcon, Franklin Lakes, NJ, USA). Transwell inserts with a fully differentiated Caco-2 monolayer were added into the transwell plate pre-loaded with macrophages isolated from PBMCs of CD patients. We obtained these groups of cells: Caco-2 alone and Caco-2 plus CD macrophages treated or not with JWH-133 [100 nM] and AM630 [10 μM] for 48 h. The supernatants of Caco-2 cells were collected to analyze IL-6 and IL-15 release and cells were used to perform cell viability assay. 

### 2.5. Drugs and Treatments

Macrophages were treated with JWH-133 (potent CB2 selective agonist) and with AM630 (CB2 inverse agonist). JWH-133 and AM630 (Tocris, Avonmouth, UK) were dissolved in PBS containing dimethyl sulfoxide (DMSO). DMSO final concentration on cultures was 0.01%. Macrophages were treated with JWH-133 [100 nM] and AM630 [10μM] for 48 h. For the same treatment time, non-treated cultured cells have been maintained in incubation media with or without vehicle (DMSO 0.01%). The concentrations of the drugs were defined through concentration-response experiments and were those producing the strongest effect without altering cells viability.

### 2.6. Protein Isolation, Western Blot

Proteins were isolated from macrophages cultures through RIPA lysis and extraction buffer (Millipore, Italia), according to the manufacturer’s instructions. To quantify the protein concentration, the Bradford dye-binding method has been used (Bio-Rad, Hercules, CA, USA). CB2, CCR7, DMT1, CD206, pSTAT6 and FPN-1 proteins were revealed in the macrophages’ total lysates by Western Blotting. Fifteen micrograms of denatured protein was loaded. Membranes were incubated overnight at 4 °C with rabbit polyclonal anti-CB2 (1:500 dilution; Elabscience rabbit polyclonal anti-CCR7 (1:550 dilution; Elabscience, Houston, TX, USA), mouse monoclonal anti-DMT1 (1:100 diluition; Santa Cruz, Santa Cruz, CA, USA), mouse monoclonal anti-CD206 (1:250 dilution; Santa Cruz), rabbit polyclonal anti-pSTAT6 (1:500 dilution; Elabscience, Houston, TX, USA), rabbit polyclonal anti-FPN-1 (1:1000 dilution; Novus Biologicals LLC, Centennial, Colorado) and then with the relative secondary antibody for 1 h. Reactive bands were detected by chemiluminescence (Immobilion Western Millipore) on a C-DiGit blot scanner (LI-COR Biosciences, Lincoln, NE, USA). A mouse monoclonal anti-β-Actin antibody (1:100 dilution; Santa Cruz, Santa Cruz, CA, USA) was used as housekeeping protein to verify the protein loading. Images were captured, stored and analyzed using “Image studio Digits ver. 5.0” software.

### 2.7. ELISA

ELISA Assays were performed to define IL-6, TNF-α, IL-10, IL-15 and Hepcidin concentration in the macrophage culture supernatants, using several Human ELISA Kits (Invitrogen by Thermo Fisher, Waltham, MA, USA). Standards and supernatants were pipetted into the wells of the microplate (coated with monoclonal antibodies specific to the cytokines) and were loaded in duplicate. After several plate washes, enzyme-linked polyclonal antibodies specific for IL-6, TNF-α, IL-10, IL-15 and Hepcidin were added to the wells. The addition of a substrate solution allowed us to detect the reaction. The optical density was measured at 450 nm through the Tecan Infinite M200 (Tecan Group Ltd., Männedorf, Switzerland) spectrophotometer. Cytokine’s concentrations (pg/mL) were determined against a standard concentration curve.

### 2.8. Iron Assay

After 48 h treatment, macrophage culture supernatants were collected to quantify the iron (III) concentration. The assay was performed using the Iron Assay Kit (Abcam, Cambridge, UK) and the macrophage supernatants were pipetted into the wells and were incubated with an acidic buffer to allow iron release. Then, an iron probe at 25 °C for 60 min in the dark was added. Released iron reacted with the chromogen producing a colorimetric (593 nm) product, proportional to the iron concentration. The optical density was measured at 593 nm through the Tecan Infinite M200 (Tecan Group Ltd., Männedorf, Switzerland) spectrophotometer. Iron (II) and total iron (II + III) contents of the test samples (nmol/μL) were determined against a standard concentration curve. Iron (III) content can be calculated as: Iron (III) = Total Iron (II + III) − Iron (II).

### 2.9. Count and Viability Assay Kit

Caco-2 cells were isolated, after 48 h of drug exposure, from the coculture media to perform the count and viability assay, with the Muse cell analyzer machine and with the Count & Viability Assay Kit. The Muse Count & Viability reagent differentially stains viable and non-viable Caco-2 cells based on their permeability to the two DNA-binding dyes present in the reagent. In total, 450 µL of the Muse Count & Viability reagent has been mixed to 50 µL a Caco-2 suspension (1 × 10^5^ cells/mL) and incubated for 5 min at room temperature. The results were analyzed with Muse 1.4 analysis software.

### 2.10. Statistical Analysis

Results are expressed as means ± standard deviation (SD). Data were obtained from independent experiments on each individual sample. Statistical analyses on data were performed using the Student’s *t*-test (XLSTAT by Addinsoft 2020. Boston, MA, USA) to evaluate differences between quantitative variables. Data are expressed as mean ± SD. A *p* value ≤ 0.05 (*) or (^) was considered statistically significant.

## 3. Results

### 3.1. Characterization of Macrophages Derived from CD Patients

Firstly, we performed a Western blot (WB) to evaluate the protein expression levels of CB2 in macrophages isolated from the peripheral blood of CD patients. Biochemical analysis revealed that, in CD macrophages, CB2 was significantly lower than CTR macrophages (Figure 1A). Then, to investigate CD macrophages phenotype we evaluated the protein expression levels of M1 polarization markers, CCR7 and DMT1 (Figure 1B,C), and of the M2 polarization markers, CD206 and pSTAT6 (Figure 1D,E). Biochemical analysis revealed that, in CD macrophages, the levels of M1 and M2 polarization markers are, respectively, higher and lower compared with CTR macrophages, suggesting that, in CD patients, there was a prevalence of the inflammatory M1 macrophage phenotype. Moreover, we also performed several enzyme-linked immunosorbent assays (ELISA) to evaluate cytokine release by CD macrophages compared to CTR macrophages (Figure 2). Confirming the existence of a M1 phenotype, the results of ELISA revealed a significant increase in pro-inflammatory cytokines, IL-6 and TNF-alpha (Figure 2A,B), and a reduction in the anti-inflammatory cytokine, IL-10 (Figure 2C).

### 3.2. Evaluation of Iron Metabolism in Macrophages Derived from CD Patients

To evaluate iron metabolism in macrophages derived from CD patients, we measured hepcidin levels, the intracellular ferric iron ion concentration [Fe^3+^] and FPN-1 protein expression (Figure 3). Hepcidin, the main factor responsible for iron metabolism modulation, acts by binding to the iron transporter ferroportin 1 (FPN-1) causing its internalization and degradation [26]. IL-6 is the principal mediator of hepcidin up-regulation [37]. In agreement with IL-6 increase in CD macrophages, we observed an increase in hepcidin levels and a consequent reduction in FPN-1 protein expression, compared to CTR macrophages (Figure 3A,C). Accordingly, we observed an increase in [Fe^3+^] in CD macrophages (Figure 3B) further suggesting the inflammatory state of these macrophages. These data also agree with the increase in the iron transporter DMT1 found in CD macrophages (Figure 1C).

### 3.3. Effects of CB2 Modulation on CD Macrophage Polarization

To investigate the role of CB2 receptor on CD macrophage polarization, we treated in vitro CD macrophages with the CB2 agonist JWH-133 [100 nM] and the CB2 inverse agonist, AM630 [10 µM] for 48 h. We analyzed the effect on macrophage marker expression and cytokine release through WB and ELISA assays. The administration of JWH-133 [100 nM] determined a significant reduction in CCR7 and DMT1 and a concomitant increase in M2 markers, CD206 and pSTAT6 inducing a polarization toward the anti-inflammatory M2 phenotype (Figure 4). After CB2 blockade with AM630 [10 µM], we observed, as expected, an opposite trend compared to JWH-133 administration and even an aggravation of the inflammatory condition compared to NT. We did not observe any significant changes for CCR7, CD206 and pSTAT6 markers (Figure 4A,C,D), while we found a significant increase in DMT1 protein expression (Figure 4B). Accordingly, ELISA assays revealed a significant reduction in IL-6 and TNF-alpha and an increase in the anti-inflammatory cytokine IL-10 after JWH-133 administration (Figure 5). Conversely, AM630 administration induced a significant increase in IL-6 release thus worsening the basal condition, and no significant changes in TNF- alpha and IL-10 release, leaving the condition unchanged compared to the not treated (NT) (Figure 5).

### 3.4. Effects of CB2 Modulation on CD Macrophages’ Iron Metabolism

To investigate the role of CB2 receptor on CD macrophages’ iron metabolism, we analyzed the effect on Hepcidin release and FPN-1 protein expression through WB and ELISA assays after 48 h JWH-133 [100 nM] and AM630 [10 µM] administration. The administration of JWH-133 [100 nM] determined a significant reduction in Hepcidin release and a concomitant significant increase in the only known exporter of iron, FPN-1. The strong increase in FPN-1 observed following CB2 stimulation could have been mediated by the reduction in hepcidin release and by a direct effect of the drug on FPN-1. Conversely, after the CB2 blockade with AM630, we observed no significant changes in Hepcidin release and a reduction in FPN-1, worsening the basal condition (Figure 6). These data further confirm the hypothesis of a direct effect of the receptor on FPN-1.

### 3.5. Effect of CB2 Modulation on IL-6 and IL-15 Release by Caco-2 Cells Alone and in Coculture with CD Macrophages

To investigate the effects of the pharmacological modulation of the CB2 receptor on intestinal barrier functionality, we cocultured CD macrophages with Caco-2 cells, widely used as a model of the intestinal epithelial barrier. We treated this model of “immunocompetent gut” with the CB2 agonist JWH-133 [100 nM] and the CB2 inverse agonist, AM630 [10 µM] for 48 h and analyzed their effects on IL-6 and IL-15 release by ELISA assay (Figure 7). We observed a significant increase in IL-6 and IL-15 released by Caco-2 cells cocultured with untreated CD macrophages compared to Caco-2 cells alone, suggesting a contribution of CD M1 macrophages in inducing epithelium inflammation. Interestingly, after JWH-133 [100 nM] administration we observed a reduction in both cytokine release. As expected, the inverse agonist, AM630 [10 µM] increased the release of pro-inflammatory cytokines by Caco-2 cells, worsening the basal condition (Caco-2 alone).

### 3.6. Effect of CB2 Modulation on Caco-2 Viability Alone and in Coculture with CD Macrophages

We also analyzed the viability of Caco-2 cells, alone and cocultured with CD macrophages, treated or not with JWH-133 [100 nM] and AM630 [10 µM] for 48 h (Figure 7). We observed a reduction in the Caco-2 cells’ viability when they were cocultured with CD M1 macrophages, suggesting a role of CD M1 macrophages in damaging epithelium. JWH-133 [100 nM] administration was able to restore Caco-2 viability increasing the number of viable Caco-2 cells. The inverse agonist, AM630 [10 µM], blocking CB2 receptor, induced the opposite effect reducing the viable cell number that was comparable to the Caco-2 cell number in coculture with CD M1 macrophages.

## 4. Discussion

Recent studies suggest that macrophages contribute to CD pathogenesis [5,6]. Novel possible therapeutic strategies for CD could aim to switch the macrophages from M1 to M2 phenotype reverting the typical chronic inflammatory immune response of CD. Therefore, we isolated primary macrophages from the peripheral blood of CD patients and analyzed the possibility to induce a macrophage phenotype switch towards M2, by Cannabinoid Receptor 2 (CB2) selective modulation. CB2 has important anti-inflammatory and immunoregulatory properties modulating T helper cells and inhibiting pro-inflammatory cytokine production [29,30]. In agreement with the well-known protective role of the receptor in inflammatory and immune regulation, in CD macrophages, CB2 was significantly lower than in CTR macrophages. Accordingly, we found an increase in M1 markers, CCR7 and DMT1, and a concomitant reduction in M2 markers, CD206 and pSTAT6, in CD macrophages compared to those isolated from healthy donors. CCR7 and CD206 are among the most used markers to distinguish M1 and M2 phenotypes, respectively [17,20]. Moreover, several members of the STAT family are involved in regulating macrophage phenotype. STAT6 phosphorylation is related to M2 activation [23]. Confirming the prevalence of M1 phenotype in CD, we observed an increased release of pro-inflammatory cytokines, IL-6 and TNF-alpha, together with a reduction in the anti-inflammatory cytokine, IL-10. Furthermore, CD macrophages showed an increase in the iron uptake transporter DMT1, and accordingly, an enhanced iron release. This result confirmed the inflammatory profile of CD macrophages, indeed the M1 phenotype is known to sequester intracellular iron [24,25]. Studies by Nemeth et al. have shown that hepcidin binds to ferroportin 1 (FPN-1), which is the unique iron export protein in mammalian cells, and mediates its internalization and degradation, resulting in a decrease in iron release [26]. Expression of hepcidin is induced by IL-6 and LPS, indeed during inflammation its expression increases [37]. Accordingly, we observed an increase in hepcidin release by CD macrophages and a consequent reduction in FPN-1. After treating CD macrophages with CB2 agonist, JWH-133, we observed a phenotype switch toward the anti-inflammatory and immune suppressive M2 type as demonstrated by the evaluation of the macrophage markers’ expression and by the analysis of the cytokine profile, thus confirming the anti-inflammatory and immunomodulatory role of CB2 receptor. In addition, the blockade of the receptor with the inverse agonist, AM630, induced significant opposite effects demonstrating the real involvement of CB2 in CD macrophage polarization. Our data agree with several recent findings about the role of CB2 in macrophage polarization [38,39]. Lin Li et al. demonstrated that CB2 agonism promotes protective M2 polarization in microglia inducing anti-inflammatory effects [38]. Rzeczycki et al. demonstrated, in a model of joint injury, the anti-inflammatory effects of CB2 stimulation by modulating macrophage polarization [39]. Moreover, we analyzed the effects of the selective modulation of CB2 on hepcidin release by CD macrophages and FPN-1 expression. Interestingly, the stimulation of the receptor with the agonist JWH-133 strongly reduced hepcidin levels increasing FPN-1 expression which contributes to the restoration of iron concentrations by regulating its transport, thus concurring to reduce iron-related inflammation. This effect could be a consequence of the reduction in the pro-inflammatory cytokine, IL-6, whose release is significantly lowered after JWH-133 administration, alternatively it could be linked to a direct effect on hepcidin release. Although further studies are needed to elucidate this interesting aspect, certainly, it suggests that CB2 stimulation could revert the inflammatory state related to the presence of M1 macrophages in CD. The intestinal epithelial barrier functionality is strongly altered in CD mainly due to constitutional changes. In particular, the epithelial dysfunction is related to mature epithelial cell reduction that increases intestinal permeability altering gut homeostasis [40,41]. To evaluate the contribution of M1 CD macrophages in inducing mucosal barrier damage and to confirm the protective effects of the CB2 modulation, we reproduced a model of “immunocompetent gut” [34], constituted by CD macrophages and the human epithelial cell line, Caco-2, widely used as a model of intestinal epithelial barrier. One of Caco-2 properties is the ability to spontaneously differentiate into a monolayer of cells with typical properties of absorptive enterocytes [42]. As expected, we observed an increase in IL-6 and IL-15 levels and a reduction in viable Caco-2 cells when CD macrophages and Caco-2 cells were cocultured, thus confirming the role of M1 CD macrophages in inducing mucosal barrier damage. JWH-133 administration inhibited the barrier damage induced by M1 CD macrophages as demonstrated by IL-6 and IL-15 reduction and by preventing Caco-2 cell viability. The reduction in IL-15 is interesting data considering the role of this pro-inflammatory cytokine in the generation of epithelial damage in active CD [43]. IL-15 is secreted by the intestinal epithelium and is higher in the lamina propria and the intestinal epithelium of untreated CD patients as compared with treated patients and the controls [44,45]. Moreover, IL-15 affects proliferation and function of intraepithelial lymphocytes (IEL) in the intestinal mucosa of CD patients [45]; thus, its reduction could limit the uncontrolled IEL activation and survival. Collectively, our data demonstrated CB2 receptor involvement in CD-related inflammatory state suggesting that it could be an important anti-inflammatory and immunomodulatory target. Moreover, we found an increased expression of M1 macrophages and demonstrated their involvement in inducing the typical mucosal barrier damage of CD. Interestingly, CB2 stimulation switched macrophage polarization towards the anti-inflammatory M2 phenotype thus reducing inflammation but also limiting the epithelial dysfunction. The main limitation of the present study is the small number of patients. However, this is an exploratory analysis that should be validated in larger samples and certainly, further investigations are needed to perform in vivo investigations to evaluate the effects of the pharmacological modulation of CB2 on macrophage polarization and mucosal barrier damage. Despite these limitations, the possibility to obtain samples from pediatric subjects is certainly an important and noteworthy strength of our study. The second limitation, concerning the absence of in vivo studies, could be overcome considering that our study was conducted on primary cells obtained from CD pediatric patients at diagnosis. Finally, another possible limitation could be linked to the use of macrophages isolated from the peripheral blood of CD patients rather than intestinal ones. Their use is still difficult due to the poor yield obtained by the current protocols; however, our future plan is to confirm our results from macrophages isolated from intestinal biopsies. Nevertheless, it is known that gliadin induces pro-inflammatory activation not only of intestinal macrophages but also of circulating monocytes from which macrophages are derived, making these cells a good model for studying CD pathogenesis. We also confirmed the results obtained from the macrophage cultures on a model of “immunocompetent gut” [34], further validating the use of this culture system to investigate the effectiveness of new possible therapeutic strategies for CD. 

## 5. Conclusions

In this study, we investigated the phenotype of macrophages isolated from the peripheral blood of CD patients and CB2 expression, amply known for its anti-inflammatory and immunomodulatory properties. We found an increased expression of M1 macrophages and as expected, a CB2 reduced expression. We also demonstrated CD M1 macrophage involvement in inducing the typical mucosal barrier damage of CD. Confirming its important properties, CB2 stimulation switches macrophage polarization towards the anti-inflammatory M2 phenotype, thus reducing inflammation and limiting the epithelial dysfunction. Taken together, our data indicate that the CB2 receptor is a possible novel therapeutic target for CD, through the regulation of macrophage polarization and the prevention of mucosal barrier damage. 

## Figures and Tables

**Figure 1 biomedicines-10-00874-f001:**
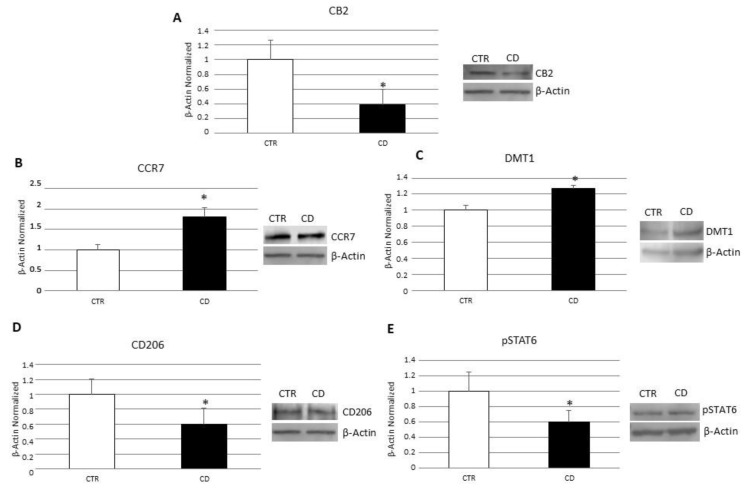
Characterization of CD macrophages. CB2 (**A**), CCR7 (**B**), DMT1 (**C**) CD206 (**D**) and pSTAT6 (**E**) protein levels, determined by Western Blot loading 15 μg of total lysate, in macrophages from 5 celiac disease (CD) patients compared with macrophages from 5 healthy donors (CTR). The most representative images are displayed. Image Studio Digits software has been used to detect protein bands. The intensity ratios of immunoblots compared to CTR, considered 1, were quantified after normalizing with respective controls. The histograms represent the relative quantification for CB2, CCR7, DMT1, CD206 and pSTAT6 expression, normalized for the housekeeping protein β-Actin, as mean ± SD of independent experiments on each individual sample. A *t*-test has been used for statistical analysis. * Indicates *p* ≤ 0.05 compared to CTR.

**Figure 2 biomedicines-10-00874-f002:**
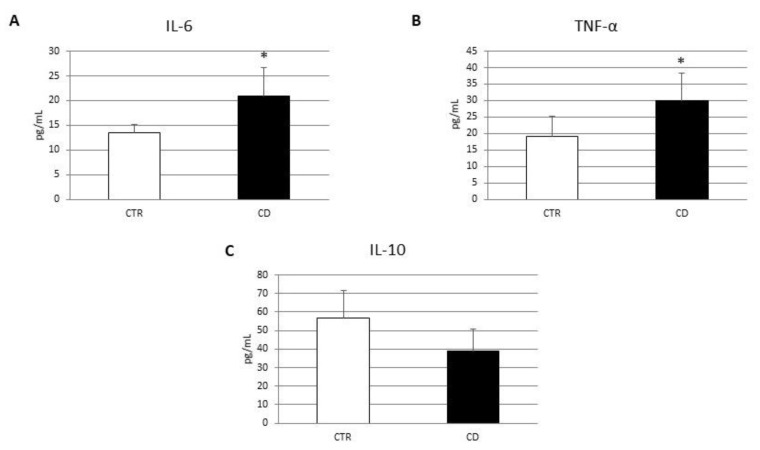
Cytokines concentration evaluation in CD macrophages. IL-6 (**A**), TNF-alpha (**B**) and IL-10 (**C**) concentrations (pg/mL) in macrophages from 5 CD patients compared with macrophages from 5 CTR, determined by ELISA Assay. Histogram shows the cytokine concentration as the mean ± S.D of different experiments on each individual sample. The concentrations of IL-6, TNF-α and IL-10 were determined on specific standard concentration curves. A *t*-test has been used for statistical analysis. * Indicates *p* ≤ 0.05 compared to CTR.

**Figure 3 biomedicines-10-00874-f003:**
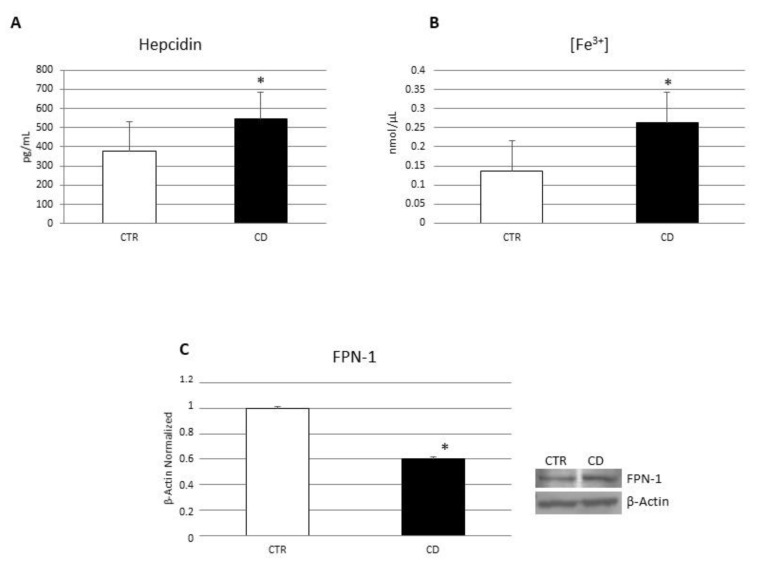
Iron metabolism evaluation in CD macrophages. (**A**) Hepcidin concentrations (pg/mL) in macrophages from 5 CD patients compared with macrophages from 5 CTR, determined by ELISA Assay. Histogram shows Hepcidin concentration as the mean ± S.D of independent experiments on each individual sample. The cytokine concentration was determined on a standard concentration curve according to the manufacturer’s instructions. (**B**) Fe3+ intracellular concentrations (nmol/µL) in macrophages from 5 CD patients compared with macrophages from 5 CTR determined by Iron Assay. Histogram shows Fe3+ concentration as the mean ± SD of independent experiments on each individual sample. (**C**) Ferroportin (FPN-1) protein levels, determined by Western Blot, loading 15 μg of total lysate, in macrophages from 5 celiac disease (CD) patients compared with macrophages from 5 healthy donors (CTR). The most representative images are displayed. Image Studio Digits software has been used to detect protein bands. The intensity ratios of immunoblots compared to CTR, considered 1, were quantified after normalizing with respective controls. The histograms represent the relative quantification for FPN-1 expression, normalized for the housekeeping protein β-Actin, as mean ± SD of independent experiments on each individual sample. A *t*-test has been used for statistical analysis. * Indicates *p* ≤ 0.05 compared to CTR.

**Figure 4 biomedicines-10-00874-f004:**
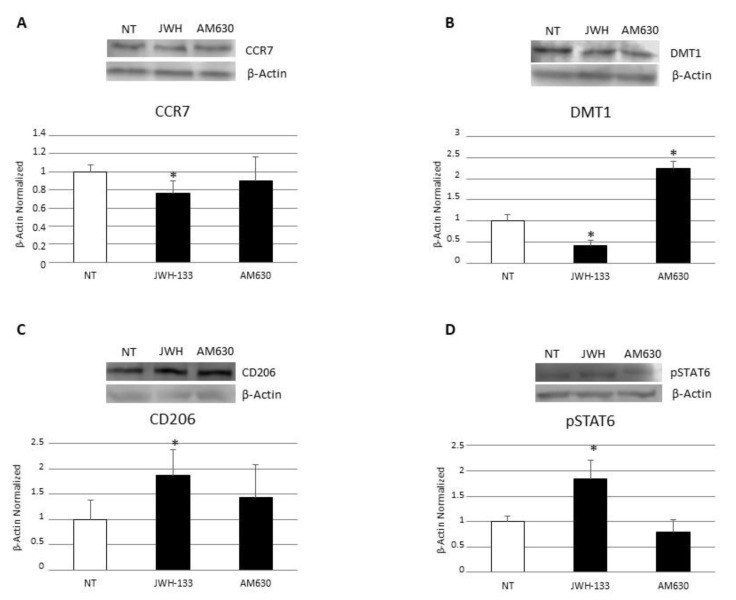
Effects of CB2 modulation on CD macrophages polarization. CCR7 (**A**), DMT1 (**B**) CD206 (**C**) and pSTAT6 (**D**) protein expressions, determined by Western Blot, loading 15 μg of total lysate, in macrophages from 5 celiac disease (CD) patients after JWH-133 [100 nM] and AM630 [10 µM] administration for 48 h. The most representative images are displayed. Image Studio Digits software has been used to detect protein bands. The intensity ratios of immunoblots compared to CTR, considered 1, were quantified after normalizing with respective controls. The histograms represent the relative quantification for CCR7, DMT1, CD206 and pSTAT6 expression, normalized for the housekeeping protein β-Actin, as mean ± SD of independent experiments on each individual sample. A *t*-test has been used for statistical analysis. * Indicates *p* ≤ 0.05 compared to NT.

**Figure 5 biomedicines-10-00874-f005:**
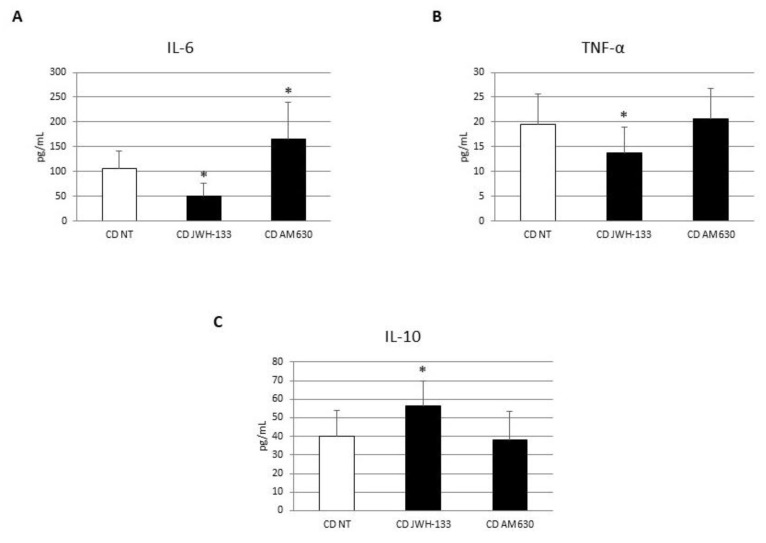
Effects of CB2 modulation on cytokines release by CD macropahges. IL-6 (**A**), TNF-alpha (**B**) and IL-10 (**C**) concentrations (pg/mL) in macrophages from 5 CD patients after 48 h JWH-133 [100 nM] and AM630 [10 µM] administration, determined by ELISA Assay. Histogram shows cytokines concentration as the mean ± S.D of different experiments on each individual sample. The concentrations of IL-6, TNF-α and IL-10 were determined on specific standard concentration curves. A *t*-test has been used for statistical analysis. * Indicates *p* ≤ 0.05 compared to NT.

**Figure 6 biomedicines-10-00874-f006:**
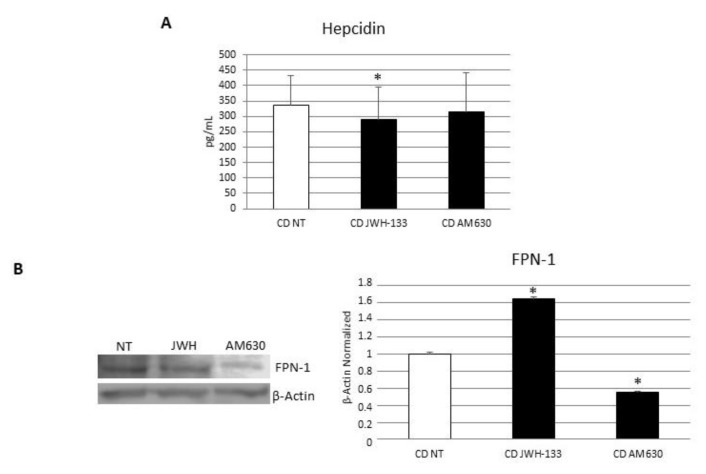
Effects of CB2 modulation on CD macrophages iron metabolism. (**A**) Hepcidin concentrations (pg/mL) in macrophages from 5 CD patients after 48 h JWH-133 [100 nM] and AM630 [10 µM] administration determined by ELISA Assay. Histogram shows Hepcidin concentration as the mean ± S.D of independent experiments on each individual sample. The cytokine concentration was determined on a standard concentration curve according to the manufacturer’s instructions. (**B**) Ferroportin (FPN-1) protein expression, determined by Western Blot, loading 15 μg of total lysate, in macrophages from 5 CD patients after JWH-133 [100 nM] and AM630 [10 µM] administration for 48 h. The most representative images are displayed. The intensity ratios of immunoblots compared to CTR, considered 1, were quantified after normalizing with respective controls. The histograms represent the relative quantification for FPN-1 expression, normalized for the housekeeping protein β-Actin, as mean ± SD of independent experiments on each individual sample. A *t*-test has been used for statistical analysis. * Indicates *p* ≤ 0.05 compared to NT.

**Figure 7 biomedicines-10-00874-f007:**
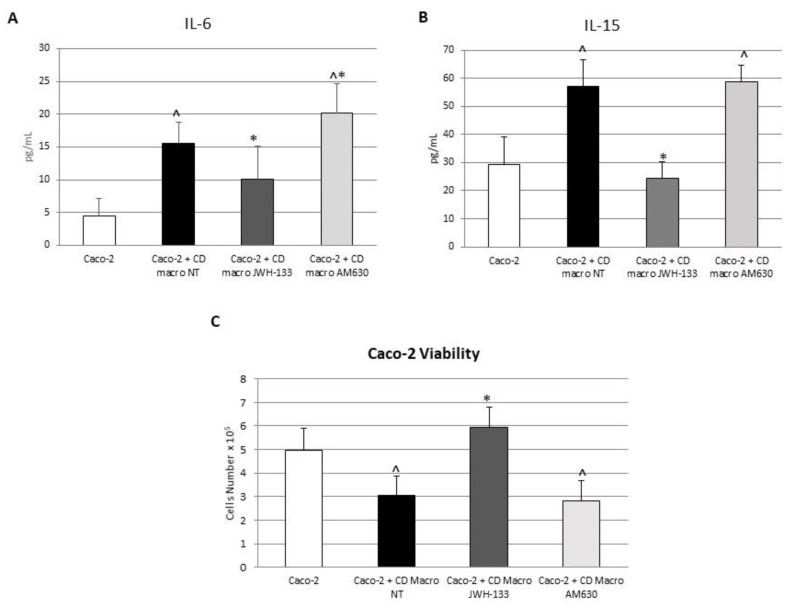
Effects of CB2 modulation on cytokines release by Caco-2 cells and on Caco-2 viability. (**A**,**B**) IL-6 and IL-15 concentrations (pg/mL) in Caco-2 cells alone and in coculture with CD macrophages isolated from 5 CD patients after 48 h JWH-133 [100 nM] and AM630 [10 µM] administration, determined by ELISA Assay. Histogram shows cytokine concentration as the mean ± S.D of independent experiments on each individual sample. The cytokine concentration was determined on a standard concentration curve according to the manufacturer’s instructions. (**C**) Caco-2 cell viability. The viability of Caco-2 cells cocultured with CD macrophages was estimated by a cytofluorimetric assay after 48 h treatment with JWH-133 [100 nM] and AM630 [10 µM]. A *t*-test has been used for statistical analysis. ^ indicates *p* ≤ 0.05 compared to Caco-2; * indicates *p* ≤ 0.05 compared to Caco-2 + CD macrophages NT.

**Table 1 biomedicines-10-00874-t001:** Clinical characteristics of the controls (CTR) and celiac disease (CD) subjects.

Clinical Characteristics	CTR	CD
**Median age, years (mean ± SD)**	12 ± 5	10 ± 4
**Sex (Female/Male)**	5/5	6/4
**Sideremia (µg/dL)**	87.1 ± 33.9	78.4 ± 25.9
**Ferritin (ng/mL)**	31.63 ± 20	19.5 ± 10
**Transferrin (mg/dL)**	256.5 ± 14	292.5 ± 34.5
**C-Reactive Protein (mg/L)**	0.08 ± 0.18	0.08 ± 0.11
**Hemoglobin (g/dL)**	13.3 ± 0.99	12.9 ± 0.89
**Mean Corpuscular Volume fl**	79.7 ± 3.6	77.2 ± 6.6
**Transferrin Saturation Index (%)**	30.5 ± 10	18.4 ± 6.5

This table shows the clinical characteristics of CTR and CD subjects enrolled in the study (*n* = 10).

## Data Availability

The raw data supporting this article will be made available by the authors without reservation.

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
