# Peer review of "Effects of CB2 Receptor Modulation on Macrophage Polarization in Pediatric Celiac Disease"

_biomedicines, 2022, doi:10.3390/biomedicines10040874_

Round 1

Reviewer 1 Report

This manuscript is well-written, with attention to details and with huge potential impact on therapy of patients with celiac disease. It appears well organized and data are of high quality. I truly appreciate the authors’ ideas for their study. Methods and Results are clearly and concisely described and explained. Discussion paragraph is nicely conceived, including comparison with other studies from the literature. References are well chosen and up-to-date. Minor suggestions/comments:

  1. Abstract: a. Line 16 - Please correct: Celiac disease represents an autoimmune disease/disorder, not just “immune-mediated enteropathy”. b. Please also add “in genetically susceptible individuals” after “gluten”. c. Please emphasize clearer and more concisely the aims of your study. d. Please insert the period the study was performed.
  2. Keywords: It would be advisable to use other Keywords, not (all) those belonging to the title. This would increase the likelihood of the paper being found by readers. The importance of Keywords is to improve indexing.
  3. Introduction: lines 35-36 – same remarks as in the Abstract. Otherwise, very well organized, clear and introducing the importance of the study. Aims appear better formulated here.
  4. Materials and Methods: a. 2.1 Please write “Patients” and include the duration of your study. b. Line 82: please insert “.” (period). c. Line 88: Please revise “not assumption of nutritional integration.” and make it clearer. What does it belong to? d. I would also suggest to include in table 1 the characteristics of the healthy controls.
  5. Results: I wonder, why in Figures, the authors chose to show only data from 5 CD patients and 5 healthy controls and not the entire groups (each made of 10 persons). Otherwise, Figures are of good quality and illustrative. Figure legends are clearly described.
  6. Please insert in “Discussion” the strength and limitations of your study. This is of paramount importance.
  7. Please insert a more generous conclusion, including also practical implications of your results and directions for future research.
  8. Please ensure that all references have the right format.

Author Response

Reviewer 1

This manuscript is well-written, with attention to details and with huge potential impact on therapy of patients with celiac disease. It appears well organized and data are of high quality. I truly appreciate the authors’ ideas for their study. Methods and Results are clearly and concisely described and explained. Discussion paragraph is nicely conceived, including comparison with other studies from the literature. References are well chosen and up-to-date. Minor suggestions/comments:

-Abstract: a. Line 16 - Please correct: Celiac disease represents an autoimmune disease/disorder, not just “immune-mediated enteropathy”. b. Please also add “in genetically susceptible individuals” after “gluten”. c. Please emphasize clearer and more concisely the aims of your study. d. Please insert the period the study was performed.

Response

We are very thankful for the comments and for the opportunity to improve our manuscript. As requested, we revised abstract section emphasizing the aims of our study.

-Keywords: It would be advisable to use other Keywords, not (all) those belonging to the title. This would increase the likelihood of the paper being found by readers. The importance of Keywords is to improve indexing.

Response

As requested, we added other keywords.

-Introduction: lines 35-36 – same remarks as in the Abstract. Otherwise, very well organized, clear and introducing the importance of the study. Aims appear better formulated here.

Response

As requested, we revised lines 35-36 of introduction section.

-Materials and Methods: a. 2.1 Please write “Patients” and include the duration of your study. b. Line 82: please insert “.” (period). c. Line 88: Please revise “not assumption of nutritional integration.” and make it clearer. What does it belong to? d. I would also suggest to include in table 1 the characteristics of the healthy controls.

Response

As requested in the revised version of our manuscript we included the duration of our study and we added in table 1 the characteristics of the healthy controls in Materials and Methods section.

-Results: I wonder, why in Figures, the authors chose to show only data from 5 CD patients and 5 healthy controls and not the entire groups (each made of 10 persons). Otherwise, Figures are of good quality and illustrative. Figure legends are clearly described.

Response

Although working with patients is our greatest advantage, at the same time it is our biggest limitation because the quality and the quantity of the samples obtained does not always allow us to use all the samples for all the experiments (Western blot, Iron assay, Elisa assay, co-cultures). As suggested by the Reviewer in the point 6, in the revised version of our manuscript, we discussed the strength and the limitations of our study.

-Please insert in “Discussion” the strength and limitations of your study. This is of paramount importance.

Response

As requested, we added the strength and limitations of our study in discussion section.

-Please insert a more generous conclusion, including also practical implications of your results and directions for future research.

Response

As requested, we added a generous conclusion section.

-Please ensure that all references have the right format.

Response

We checked the references format.

Reviewer 2 Report

A very interesting original in vitro study evaluating the effects of cannabinoid receptor 2 stimulation to macrophage polarization, noticing a switch of the polarization to the anti-inflammatory phenotype, reducing inflammation, and limiting epithelial disfunction.

Only minor queries: 

line 40 you should add: to further enhance this hypothesis, various macrophage-initiating granulomatous conditions have been associated with CD" and cite a paper such as: doi: 10.3390/medicina55090578.

Thank You!

Author Response

Reviewer 2

A very interesting original in vitro study evaluating the effects of cannabinoid receptor 2 stimulation to macrophage polarization, noticing a switch of the polarization to the anti-inflammatory phenotype, reducing inflammation, and limiting epithelial disfunction.

Only minor queries: 

line 40 you should add: to further enhance this hypothesis, various macrophage-initiating granulomatous conditions have been associated with CD" and cite a paper such as: doi: 10.3390/medicina55090578.

Thank You!

Response

We are very thankful for the comments and for the opportunity to improve our manuscript. In the revised version of our manuscript, we added the sentence “various macrophage-initiating granulomatous conditions have been associated with CD" citing the paper suggested
